# Integrated Information Theory and the Phenomenal Binding Problem: Challenges and Solutions in a Dynamic Framework

**DOI:** 10.3390/e27040338

**Published:** 2025-03-25

**Authors:** Chris Percy, Andrés Gómez-Emilsson

**Affiliations:** 1College of Arts, Humanities and Education, University of Derby, Derby DE22 1GB, UK; 2Qualia Research Institute, San Francisco, CA 95066, USA; algekalipso@gmail.com

**Keywords:** integrated information theory, phenomenal consciousness, binding problem, phenomenal binding, temporal dynamics

## Abstract

Theories of consciousness grounded in neuroscience must explain the phenomenal binding problem, e.g., how micro-units of information are combined to create the macro-scale conscious experience common to human phenomenology. An example is how single ‘pixels’ of a visual scene are experienced as a single holistic image in the ‘mind’s eye’, rather than as individual, separate, and massively parallel experiences, corresponding perhaps to individual neuron activations, neural ensembles, or foveal saccades, any of which could conceivably deliver identical functionality from an information processing point of view. There are multiple contested candidate solutions to the phenomenal binding problem. This paper explores how the metaphysical infrastructure of Integrated Information Theory (IIT) v4.0 can provide a distinctive solution. The solution—that particular entities aggregable from multiple units (‘complexes’) define existence—might work in a static picture, but introduces issues in a dynamic system. We ask what happens to our phenomenal self as the main complex moves around a biological neural network. Our account of conscious entities developing through time leads to an apparent dilemma for IIT theorists between non-local entity transitions and contiguous selves: the ‘dynamic entity evolution problem’. As well as specifying the dilemma, we describe three ways IIT might dissolve the dilemma before it gains traction. Clarifying IIT’s position on the phenomenal binding problem, potentially underpinned with novel empirical or theoretical research, helps researchers understand IIT and assess its plausibility. We see our paper as contributing to IIT’s current research emphasis on the shift from static to dynamic analysis.

## 1. Introduction

As you read this paper, you are able to hold multiple words in your mind simultaneously. Those words co-exist in your phenomenal experience. This is an example of a broader phenomenon called the phenomenal binding problem, which must be accounted for by theories of consciousness.

The phenomenal binding problem, as defined for this paper, is identifying the mechanism(s) which integrate micro-units of *information* into the simultaneous and informationally complex macro-scale conscious *experiences* common to human phenomenology (references in Section 2.1). An example of this problem is how single ‘pixels’ of a visual scene are experienced as a single holistic image in the ‘mind’s eye’, rather than as individual, separate, and massively parallel experiences, corresponding perhaps to individual neuron activations, neural ensembles, or foveal saccades, any of which could conceivably deliver identical functionality from an information processing point of view.

There are multiple contested candidate solutions to the phenomenal binding problem, with different positions available in computational neuroscience, philosophy of mind, and metaphysics, albeit with few explicitly developing a complete account (see Section 2.1). This paper explores how the metaphysical infrastructure of Integrated Information Theory (IIT) v4.0 can provide a distinctive candidate solution to the phenomenal binding problem, albeit not normally framed in those terms. We acknowledge that the various metaphysical claims in IIT v4.0 contain tensions that permit different interpretations [1] and motivate a specific synthesis that addresses phenomenal binding.

IIT’s solution, as we describe it, potentially provides a strong motivation for those who see causality as central to phenomenal consciousness to move away from computational functionalism towards IIT’s causal structure approach. However, IIT’s solution does not come for free. As we develop a dynamic perspective on how conscious entities and experiences develop through time, IIT is faced with five options for aligning the theory with various observations. Clarifying IIT’s position, potentially underpinned with novel empirical or theoretical research, will help researchers understand IIT and their grounds for accepting or rejecting it.

### 1.1. Paper Contributions

This paper makes four primary contributions. First, we make explicit the current way in which IIT can address the phenomenal binding problem from a static perspective, motivating an accessible synthesis of the problem and connecting IIT’s terminology with the key literature. Second, we define a comprehensive option space for IIT’s mathematical assessment of what happens to experiencing entities between one timeframe and the next, specifying the dilemma faced by IIT in interpreting those options. Third, we explore the feasibility of IIT dissolving the dilemma by rejecting closed individualism, adopting a different metaphysics, or denying the phenomenal binding problem. Fourthly, we summarise the five options by which IIT might resolve these issues, suggesting research activity that might motivate particular choices.

We see our paper as supporting the current research priority within IIT to explore dynamic implications of the theory (e.g., [2], p. 6 on temporal phenomenality), noting that early research focused on a static evaluation of transition probability matrices and causal graphs. We do not assume prior knowledge of IIT’s terminology or its metaphysics; IIT experts may wish to skip Section 2.4, Section 2.5, Section 2.6, Section 3.1 and Section 3.2.

### 1.2. Paper Summary and Section Outline

A first approximation of our argument can be presented via a section outline. Section 2 first presents a synthesis of the phenomenal binding problem, explaining how the problem remains valid even for theorists who reject an overall unity of phenomenal consciousness. Section 2 then draws together the key strands of IIT’s metaphysics that empower a solution, notably the mereological assumption that particular entities aggregable from multiple units (‘complexes’) define existence for experiencing subjects (i.e., selves) at a given moment in time, where the contents of their experience in that moment is defined by the unfolding of the complex’s Φ-structure. Such complexes encompass compositional complexity by definition and their constituent parts cease to exist while the complex persists.

Section 3 provides a diagrammatic illustration of all mathematically possible transitions for the main complex in an abstract substrate graph from one time step to the next. Of particular interest is the transition in which a separate, spatially non-contiguous complex grows to have larger φ than the previous main complex. IIT has a choice to make regarding how this transition should be interpreted, which can be presented as a dilemma. Under one horn of the dilemma, the original self remains in the area contiguous with the original substructure, with a new, separate self developing in the new substructure, implying that human consciousness might not always be the largest consciousness in a biological brain. In horn two, the self moves to the new, larger φ substructure, but must do so in a non-local fashion, perhaps defendable via intuitions of non-locality in quantum mechanics which would require specifying in a future IIT.

Section 4 describes three approaches for dissolving the dilemma, without having to accept any of the (potentially) disquieting implications of the two horns described in Section 3. The first denies the validity of introspective evidence of short-term dynamic persistence, perhaps embracing empty individualism instead (e.g., where selves do not endure over time [3]). The second dissolves the dilemma by moving away from the metaphysical solution by which IIT addresses the binding problem. However, three possibilities identified for an alternative metaphysics all appear to denude human-level consciousness of causal relevance, contradicting IIT’s present axioms. The third approach is to deny the need for phenomenal binding in the first place.

Section 5 concludes by summarising the paper into five options presented in a table and suggesting future research avenues which might make IIT’s choices easier to motivate.

### 1.3. Implications for IIT

The purpose of our paper is not to argue for one particular interpretation that IIT should adopt or to imply that all of the options we present are so unpalatable that IIT should be rejected as a theory. After all, IIT has an admirable history of identifying and then embracing seemingly counterintuitive implications of its theory (see Section 5)—it is equally feasible to bite one or more of the bullets we present in this paper. More importantly, there are further theoretical and empirical research avenues to pursue which might make particular choices easier to motivate. Instead, our purpose is to highlight this issue as deserving attention while theorists explore the implications and applications of IIT in a dynamic setting and to set out a landscape of available options to support them in doing so.

Some IIT theorists may feel that one of the presented options is so obvious that the others can be happily ignored. However, in our conversations at conferences in 2024 it does not seem that all theorists necessarily appreciate the consequences of different choices or necessarily agree about which option to pick. Indeed, it is possible that IIT advocates may diverge and create different theory variants in response to the phenomenal binding problem, similar to the divergence between weak and strong IIT [4]. In any case, explicit positions on this issue will bring useful clarity to the field.

## 2. IIT’s Solution to the Phenomenal Binding Problem

### 2.1. The Phenomenal Binding Problem

The target of this paper is the phenomenal binding problem, which can be elucidated using the “many notions of binding” that [5] describes in her widely cited paper on binding. Our presentation of the phenomenal binding problem can be mapped to the subject combination problem in [6]. While [6] targets panpsychist theories only, ref. [7] has shown the problem applies also to other theories of consciousness. In addition to citations in this section, the binding problem is discussed widely elsewhere in the context of experiential binding, e.g., [8,9,10,11].

A classic entry point is visual binding. For our visual system to function usefully, multiple pieces of information that are processed separately in the brain must be connected to each other, such as the space and location of an object. The brain may commonly succeed, but it can also be tricked into binding information in a way that does not line up with the external environment. Ref. [5] explains that such perceptual binding problems encompass binding across visual space, across features, and across cortical areas, applying both within and across different sensory modalities. She goes on to describe various cognitive binding problems and the possibility that different mechanisms address different types of binding.

All of these problems are functional in nature. They are to do with how information is processed. The mark of success for this kind of binding mechanism is whether the system takes actions in some appropriate manner, having correctly connected concepts and bits of information. In turn, many functional solutions have been proposed at different levels of analysis primarily in the fields of neuroscience and computer science, even if the mix of solutions remains contested to the present day.

Candidate examples of computational infrastructures for functional binding mechanisms include hyperplane frameworks [12], dynamic binding via reciprocal connections and re-entrant loops [13], compositional AI via grounded symbol-like representation [14], Bayesian binding [15], spatial location maps [16], and vector multiplication [17]. Other solutions point to physical features in the brain (e.g., firing rate [18]; oscillations [19]; electromagnetic fields [8]). These approaches often build on the strengths and weaknesses of early approaches, such as hierarchical cell/module specialisation or spatial convergence (e.g., [20]) or temporal synchrony (e.g., [21]). Further detail can be found in reviews such as [22,23].

Ref. [5] closes by introducing a different type of binding problem: the problem of consciousness as “perhaps the most mystifying binding problem of all”. Later researchers have built on this concept of multiple notions of binding to enrich the vocabulary, differentiating phenomenal binding from functional binding and arguing that the two problems require “very different treatment” [24] (p. 381). On Garson’s definition, phenomenal binding is how we “experience a single world rather than separate perceptual fields for each sensory modality”. Ref. [24] (p. 389) claims to dissolve the phenomenal binding problem by arguing that cross-modality binding can be delivered through the same functionality as within-modality binding (i.e., cortical representation, potentially through topographic maps). Garson may be correct in this point (or not), but it fails to address the key insight in his own formulation. His discussion of cross-modality binding is a subset of functional binding and subject to the same solution space and limitations he describes. The central distinction in his definition should instead be identified as the ‘experience’ corresponding to some information, rather than its functional utility for onward ‘information processing’. As an aside, we note that footnote 7 in [24] can be read as acknowledging this limitation. However, taken to its full conclusion, footnote 7 should have led the paper to focus on multi-modal vs. within-modality functional binding, without claiming to address ‘experience’ and ‘phenomenal’ binding.

The distinction has long been grasped in philosophy of mind. Ref. [25] defined the hard problem of consciousness as understanding why any function should ever give rise to experience, arguing that we can always imagine the same function occurring without experience. Such arguments can be motivated by, for instance, explanatory gap arguments [26] or the conceivability of p-zombies [27]. The phenomenal binding problem is related to, but distinct from the hard problem. If the hard problem asks why any function should be accompanied by experience, the phenomenal binding problem asks why that function should be accompanied by a single, unified, and complex experience, rather than multiple, scattered, separate experiences—and how that function relates to the specific way in which phenomenal information is bound together. For instance, where a function contains multiple components (e.g., individual nodes in a neural network or steps in an algorithm), there is an argument that any experiential mapping would be more plausible at the component-level.

If we accept this hard problem framing, it pushes phenomenal consciousness beyond empirical analysis alone but does not concede the considerable (if incomplete) contributions that empirical analysis can make to the debate. Chalmers acknowledged the need to define a set of “psychophysical principles” and/or “bridging principles” [25] (p. 210) which axiomatically assert that certain phenomena are identical to experience.

Such bridging principle axioms may be beyond the reach of empirical falsification, just as no experiment can prove with *100%* certainty that our sensory read-out from that experiment is not deceiving us in some way. However, the plausibility of axioms can still be argued for and they can still be evaluated transparently against alternative axioms, using various explicit explananda and desiderata [28]. Indeed, this paper is an example of such analysis: IIT’s definition of consciousness is taken as an axiom and we discuss different ways it might account for the explanandum of phenomenal binding, recognising that such accounts vary in their plausibility. Importantly, candidate bridging principle axioms can sometimes be combined with other axioms to make falsifiable predictions (as with IIT), supporting progress through the scientific method and adjusting confidence in underlying theoretical models as described by [29] and discussed in the context of consciousness theories by [30]—even if perfect certainty remains a chimaera.

### 2.2. Limitations on Collapsing Phenomenal Binding into Functional Binding

It is possible that the bridging principle we seek is a *specific type* of information processing, such that phenomenal binding collapses into a specific type of functional binding. This possibility is the promise of computational neuroscience avenues for solving the hard problem. However, without particular specifications or restrictions, information processing cannot be generically understood as a bridging principle for the human case. Two reasons can be briefly provided to exclude an identity relationship that incorporates all cases of functional binding.

First, information processing can be equally correctly understood as occurring simultaneously at multiple levels, yet our subjective experience in a given moment operates at a single level, so it is not possible to draw a direct, one-to-one identity between them. For instance, analysing information processing in the physical brain gives rise to different structures and different causal diagrams if we focus on quantum particle interactions, molecular behaviour, neuronal connections, neural ensembles, or brain modules. All sometimes involve at least minimal instances of functional binding to support that information processing. All of these systems process information in nested, interconnected but different ways; only some appear to map to conscious experience.

Second, without further restrictions, information processing does not stop at the boundaries of any particular neural ensemble or brain module. However, we know the brain conducts information processing of which we are not consciously aware, but which remains causally and informationally connected to that awareness.

It may or may not be possible to specify a type of information processing bridging principle (or explanatory identity; to use IIT terminology) that meets all the requirements for phenomenal consciousness, including the phenomenal binding problem. The salient point for this paper is that functional binding does not necessarily imply phenomenal binding. The former accounts for the existence of complex information processing and behaviours; the latter addresses the holistic presence of complex information within a single experience. The former can implement functionality via sequential processing; the latter requires at least phenomenal co-existence. We know the brain does both; it is not certain if it delivers both with the same mechanism(s). Indeed, IIT’s solution, the focus of this paper, does not adopt an information processing identity, but rather a metaphysical solution grounded in the nature of existence.

### 2.3. Robustness to a Rejection of Phenomenal Unity

Ref. [24] focused on phenomenal binding as our experience of a ‘single world’. Ref. [31] likewise focuses on the conscious experience as unified. Phenomenal unity has also been explored as an example for introducing mathematical structure into the discussion of features of consciousness ([32] Section 4; [33]). However, for the purposes of this paper, there is a sufficient explanatory problem as soon as any conceivably separate experiences are in fact experienced together.

Even if phenomenal binding does not create a single, neat, holistic experience in every time step, provided it creates any experiences that contain informational complexity, we require some mechanism for binding micro-units of information into complex information. For instance, even if consciousness is only unified when we turn attention to it, we still require an explanation for experiential unity when it does occur. Even if consciousness is fragmented, if you have ever held two words in your visual field and experienced them at the same time, some phenomenal binding mechanism must exist.

This framing ensures our discussion is robust to those who argue against the unity of consciousness, such as [34,35], regardless of the reader’s position on unity (or ours). Nonetheless, there is typically a close link between unity of consciousness and phenomenal binding—and both are incorporated together in IIT’s axioms.

### 2.4. IIT’s Acceptance of the Binding Problem

While the ‘phenomenal binding problem’ language is not commonplace in IIT’s canonical publications, IIT accepts the phenomenological observation that gives rise to it. Indeed, IIT makes it one of the five key axioms from which the theory is derived: the third axiom (‘integration’) observes that consciousness is unified into an irreducible experience. The idea that this experience has macrophenomenal complexity, i.e., can span more than one unit of information is embedded in various axioms. For instance, integration is described as occurring over sets of units (where each set can contain multiple bits of information) and the fifth axiom (‘composition’) describes unitary conscious experiences as ‘structured’, i.e., can contain more than one structural element.

Once they are properly understood, the axioms are intended to be “immediately and irrefutably true of every conceivable experience” [2] (p. 38), in that they are directly evidenced via introspection and supported by narrative reasoning. It is important to emphasise that experiences can still feel—and can still be usefully described as—unintegrated, fragmented, or chaotic, without contradicting the integration axiom [36]. Similarly, an experience can contain two unrelated elements (e.g., seeing a chair while listening to music), but still be a single unified experience overall in IIT’s definition. Without the two elements or various fractured elements being called to mind at the same time, it would not be possible to experience them as unrelated or fractured. In other words, the sensation of ‘fragmentation’ is itself part of the unified experience.

A further logic-based argument is occasionally put forward for unity-related claims in IIT [2] (p. 3). If we conceive an experience that is not unitary, we should conclude we are conceiving two (or more) separate experiences, each of which would be described as unitary in IIT terms. In other words, integration is true not only of any current experience you might reflect on, but of every conceivable experience. We are sympathetic to this axiom and argument, but they are at most an argument that because complex experiences exist then binding must occur somehow. It does not describe specifically how binding occurs.

### 2.5. IIT’s Metaphysical Solution to the Binding Problem

Our interpretation of IIT’s solution draws primarily on IIT v4.0 [2,37] and the IIT Wiki as updated in 2024 (https://sites.google.com/view/iit-wiki, accessed on 1 January 2025). Our interpretation is broadly consistent with the ‘non-solipsistic idealist realism’ interpretation of IIT in [1] (Section 4.3), but with different emphases given its motivation. We note also that micro units in IIT must also comply with IIT’s postulates in order to exist (we thank a reviewer for this clarification). As a result, micro units have some minimal consciousness, providing a route to resolving the concerns raised in [1] (Section 5) and [38].

We begin with IIT’s ontology of intrinsic existence, which consists of micro units and complexes constituted out of micro units or smaller scale complexes (nuanced as below), as well as their causal interactions over discrete time. ‘Micro units’ refers to whatever is the hypothetical minimum discrete finite unit of cause-effect power that always exists in exactly one of two states (binary unit), i.e., something below the atomic level and likely unknown to current physics. ‘Intrinsic’ existence refers to entities that exist at the entity-level in their own right, as opposed to ‘extrinsic’ existence which refers to entities that exist at the entity-level only in the consciousness of a separate ‘intrinsic’ entity (see also Section 2.6).

From an analytical perspective, complexes can be understood as sets of micro units. Specifically, a complex is the set of micro units with a particular mathematical property: it has maximal irreducibility out of all candidate sets of units that share at least one unit in common with the complex. The candidate sets of units are identified based on all possible spatiotemporal grains at which micro units might be aggregated into units in cause-effect structures that can be modelled via transition probability matrices (TPMs) between system states (a system state is the set of all binary units in the system at a given point in time). Complexes are non-overlapping. Every micro unit belongs to one and exactly one complex—including the possibility of single unit complexes where it has higher irreducibility as a single unit (e.g., considering its cause-effect relationship with itself) than any other candidate complex that includes it.

From an ontological perspective, however, complexes have a further distinctive metaphysical feature. IIT’s “*principle of being* states that *to be is to have cause-effect power*” and IIT’s “*principle of maximal existence* states that, when it comes to a requirement for existence, *what exists is what exists the most*” [2] (p. 11, box 2; emphasis in original). In other words, once any given physical substrate is broken up into complexes of different sizes, the micro units that constitute those complexes no longer exist. Such existence is also dynamic. As the system state changes, some of those micro units may become disaggregated from their previous complexes and come to exist again in their own right. This metaphysical assumption is also embodied in the fourth axiom/postulate (‘exclusion’), e.g., “overlapping substrates with lower φ_s_ are thus excluded from existence [relative to the complex that does exist]” [2] (p. 19).

The final part of the puzzle falls into place with certain other axioms and identities in IIT. The ‘zeroth axiom’ of IIT is that “experience exists: there is something” [2] (p. 3) and the first axiom is intrinsicality, such that “every experience is for the experiencer” [2] (p. 2), i.e., the existence of an experience is identical to there being a specific experiencer of that experience [2] (p. 2). The explanatory identity of IIT is that an experience is identical to the Φ-structure of an intrinsic entity, i.e., “experience is present for any substrate that fulfils the essential properties of existence” [2] (p. 2). In other words, every complex is its own experiencing entity and its experience is defined by particular features of its causal structure. Indeed, strictly speaking, a complex, in the sense of the experience it embodies, “does not exist as such, but exists “unfolded” into its Φ-structure, with all the causal distinctions and relations that compose it.” [2] (p. 29).

IIT’s metaphysical assumption is thus sufficient to solve the binding problem. Individual units of integrated information are bound together in a complex by virtue of their existence in a particular cause-effect structure; a complex’s constituents literally cease to exist as individual units. Such existence is identical with an experience, i.e., the binding is phenomenal as a metaphysical assumption. The boundary is defined likewise by the boundaries of the complex. All bindings and boundaries are precisely identifiable by a sequence of mathematical rules—albeit combinatorially explosive ones such that today’s computing capacity limits us to identifying complexes only for relatively simple systems.

### 2.6. Alternative Intuitions and Mereologies

IIT’s metaphysical assumption may conflict with various intuitions from classical mechanics (as well as quantum mechanics), but it is important to recognise that alternative assumptions typically bring their own conflicts with intuition. There are no free lunches in mereology, the area of philosophy studies the relationships between parts and wholes. The current field of mereology might reasonably be described by [39] as “crazyist”—as he does for consciousness, quantum mechanics, and certain other topics—in that all positions seem to contradict common sense and we are not (yet) epistemically compelled to accept any one position.

To elaborate, common sense intuitions driving classical mechanics include that objects exist when not observed, that all systems are fully determined by the product of their parts and interactions, that all interactions are local, that time flows forward at the same speed for all observers, and so on (e.g., see [40]; elaborated in [41]). These intuitions seem to be a fairly common (albeit not universal) result of our evolved intelligences operating within their evolutionary environment (and may well be limited in accuracy as a result, as argued by [42]), even if formalising their mathematics and exploiting the engineering applications is the result of generations of science, investment, and collective genius. Unfortunately, deeper insights into physics have chipped away at these intuitions. Relativity undermines the notion of absolute time. Many argue that quantum mechanics undermines locality. Even where other positions on quantum mechanics exist, they too reject one or more intuitions from classical mechanics (e.g., [43]).

In mereology and philosophy of science, these ‘classical mechanics’ intuitions translate into the related ideas of mereological nihilism (e.g., [44]) and reductive/flat physicalism (e.g., [45]). In brief, the fact that we might assemble a pattern of interacting atoms into an object we call a ‘cat’ does not mean that a cat exists above and beyond that assembly. The things that ‘really’ exist are the underlying atoms—or, more precisely, the to-be-discovered physical phenomena that constitute fundamental entities, whether they be strings, fields, or something else.

The counterintuitive implication, extrapolated from this seemingly reasonable intuition, is that the ‘cat’ itself does not exist as an intrinsic entity. Attempts to resolve this dilemma are a live area of research, as the citations in this section demonstrate. Certain clarifications are often attempted. The ‘cat’ may still exist epistemologically, i.e., it may be a useful epistemological concept in analysing systems—perhaps an essential one given our position as computationally bounded objects interacting with computationally irreducible systems (e.g., [46]). In other words, ‘cat’ still exists as a word and it can still usefully denote the familiar temporarily persistent physical assembly of example cats, albeit with unspoken assumptions around spatiotemporal thresholds (the cat changes over time) and fuzzy boundaries or coarse-grained sensors (moulting a single strand of fur does not remove its ‘catness’). As a word, ‘cat’ might also have its own physical existence in cerebro in the sense of connectionist structuralism [47] or some other nominalist or physicalist solution, so we do not need to invoke any Platonism or other non-physical realm for the word to have causal powers. Despite these clarifications and despite the force of Ship of Theseus arguments, there remains something counterintuitive about claiming the cat does not have ontological existence as an entity in its own right.

The point of briefly highlighting the mereological debate is to forestall concerns some readers might have that IIT’s metaphysics should be rejected a priori as a self-inflicted reductio ad absurdum: one in which atoms no longer intrinsically exist once they make up a larger object, in which certain neurons cease to exist in a conscious brain, and in which entities can pop in and out of existence in a way that conflicts with classical mechanics intuitions around the permanence of physical objects and their parts—or at least that entities assemble and disassemble far more dramatically with far less physical change than those classical intuitions typically permit.

Viewed in the context of available alternative mereologies, however, IIT might be a more common-sense solution as argued by [48]. Perhaps IIT tells us that the ‘cat’ does actually exist, unlike flat physicalism which argues the ‘cat’ is merely a word—a macrovariable in some beholder’s eye—and not actually an intrinsic object in its own right.

Unfortunately, IIT’s position is not quite this commonsensical. IIT would also say the ‘cat’ does not exist as a single intrinsic entity, rather the substrate which terminates at the cat’s spatiotemporal boundaries is made up of many different complexes. The whole ‘cat’ only exists as an extrinsic entity, i.e., in the consciousness of a separate intrinsic entity. One significant complex is assumed to be part of the neural network in the cat’s brain, corresponding to the cat’s phenomenal self and conscious experience at that moment in time. In informal IIT discussions and in papers (e.g., [49] p. 17), this complex is typically assumed to have the largest system φ across all the other complexes that comprise the whole ‘cat’ substrate. If we want there to be a single intrinsic entity called the ‘cat’ (i.e., not just a concept existing in some observer’s Φ-folds), this complex within the neural network is probably it. But that complex does not line up with what we mean when we invoke the word ‘cat’.

Effectively, IIT tells us that ‘existence’ is something different to the common intuition in the natural sciences around physical objects that interact with each other. ‘Existence’ is instead related to the ability to take and make a difference and it is grouped at the ‘system’ level, rather than at the ‘parts’ level. We note that IIT is not the only mereological theory that places existence at the middle-level of a hierarchy of nested physical objects (e.g., [50]). For a more detailed introduction to this ongoing mereological debate, please see [51].

## 3. IIT’s Dilemma Under Dynamic Evolution

This section first introduces the mathematical framework and graph grammar for understanding the evolution of complexes over time. We then provide an illustrative example for the three types of evolution that delimit our scenario of interest. Finally, we explain the dilemma posed by one particular type of dynamic evolution, where the main complex in one time step is not spatially adjacent to the main complex in the prior step.

### 3.1. Mathematical Framework for Complex Evolution

The features of the dynamic evolution we wish to address can be adequately captured in IIT’s abstract substrate graphs. Such graphs summarise causal relationships from the TPMs that drive calculations to identify complexes in a given substrate. The substrate graphs alone do not identify complexes, as it would be unwieldy to annotate all the transition probabilities onto the graph. For instance, a symmetric graph need not have symmetric complex distributions. Nonetheless, a directed connection between any two units indicates a non-zero probability of directed causal action in at least some of the states involving differential source unit activations. For some analyses, such as ours, it is possible to state all relevant details in a causal graph structure, which enables analytical traction on an aspect of IIT even where exhaustive mathematical calculation across all possible substructures is not computationally practical.

For our purposes, we can use a substrate graph grammar as described in Figure 1. Specifically, units can take one of two states (on or off) and can make a difference to other units as indicated via directed arrows.

All units in a given substrate are allocated to one and only one complex (there can be multiple complexes in a substrate and single unit complexes are permitted). A unit will always fall into the largest φ complex available to it. By largest φ, we are referring specifically to system-small-phi (φ_s_), which quantifies the irreducibility of the cause-effect state specified by a system, measuring to what extent a system as a whole specifies its cause and effect in a unitary way and which is not reducible to power specified by separate parts of the system. As short-hand and except where explicitly specified, we will continue referring to φ_s_ as ‘φ’ in this paper, since the other phi measures of integrated information are not needed for our discussion (e.g., distinction-phi, relation-phi, unit-phi, etc.). In any given substrate graph within an arbitrarily chosen boundary, there can be complexes of diverse φ-values.

The substrate modelled in the graph can still be affected (and still affect) units outside the graph boundary, but IIT can treat these as background conditions for pragmatic calculations. For instance, in practice, we might presume some causal relationship degrading or simplifying physical structure such that we can safely conclude any complexes making use of adjacent units not shown (i.e., those that would overlap with those inside the boundary) would not dominate the complexes identified in the boundary. The physical structure of the skull could serve such a purpose: the integrated information represented in any causal relationships between sets of neuronal units and adjacent units in the skull would be dominated by sets consisting purely of neuronal units.

The complex with the largest φ in a given graph is called the first or main complex, but numerous ‘minor’ complexes can also exist. In Figure 1, Abcd is the main complex; i and o are two separate minor complexes. All three complexes have positive φ so all three have some degree of conscious experience, but the volume of consciousness in the Abcd entity is far larger than in the i and o entities.

The following discussion does not lean on specific relationship strengths or unit states, so we simplify the presentation to lines of a single width (no strength specified; thereby supporting double-ended arrows for causal influence in both directions) and units as simple circles where an uppercase letter labels the unit but does not specify its state. We will work with an abstract substrate graph that consists of a mixture of different size complexes. Mathematically, all we require is:

A directed graph G, which is a pair G = (V, E), where

V is a finite set of units v_1_ to v_n_

E ⊆ V × V is a set of directed arrows. Each arrow e_jk_ = (v_j_,v_k_) ∈ E, ∈ ℝ_≥0_ represents a directed connection from unit v_j_ to unit v_k_ if it has a positive value or the absence of a connection if it has value 0.

V is partitioned into a collection of disjoint subsets termed complexes c_1_…c_m_ ∈ C, such that

Each subset c_i_ ≠ ∅ (complexes are non-empty)

c_i_∩c_j_ = ∅ ∀ i ≠ j (complexes are disjoint)

${⋃}_{\left\{i=1\right\}}^{m}c_{i}=V$ (complexes provide full coverage of units)

Singleton and equality subsets are permitted

Each complex has a φ_s_ value, c_iφ_ identified by a mathematical function: φ(C):φ:{c_1_,…,c_m_} → ∈ ℝc_iφ_ = φ(c_i_)

Let the complex c_i_ with the largest φ_s_ be termed the main complex, c_main_, assuming an adequate tie breaking process in place (e.g., as in [2] S1 Text):$$c_{main}=argmax_{c_{i}}\varphi \left(c_{i}\right)$$

Figure 2 provides a simple illustration with diverse complexes where v_1_ to v_19_ are represented by units A–S and the graph grammar follows the simplified application of Figure 1 described above, with the blue area delineating the main complex, where its φ is the largest in the given system, alongside a number of minor complexes of different sizes.

### 3.2. Dynamic Evolution in Our Scenario of Interest

The motivation for this paper is what happens to the main complex in a given causally interconnected system over time. It is important to emphasise that the φ-values of candidate complexes—and hence the location and extent of the main complex—can change dramatically given only minor changes in the system. Such changes are not limited to which units and causal connections actually exist in the substrate, but also include the system state, holding the causal structure fixed. In other words, from a biological neural network perspective, we can keep all neurons and their connections fixed but alter only which neurons are firing at a given point in time. In the context of Figure 2, we might imagine the top row units A–E as feedforward neurons, perhaps simplistic versions of single pixels that activate in response to an external light signal. Depending on those external light signals, different complexes can form across units F–S, even with no other changes in the physical structure of the neural network.

Even working just with the graph details represented in Figure 2, there are many possible differences that might be relevant for the main complex between two adjacent points in time. The main complex might expand, shrink, or stay the same in terms of the number of units constituting it. The specific units constituting it can change to various subsets in the graph, including collapsing down to a singleton set of one unit. In any of those circumstances, the main complex φ-value might increase, reduce (perhaps all the way to the ‘background noise’ of adjacent complexes), or stay the same, although this would not be visible on the graph, beyond it still being identifiable via the blue shading that it is the complex with highest φ-value out of those shown. Even where the main complex stays the same, with the same units and the same φ-value, its conscious experience might nonetheless change given other changes in the system state, noting that multiple Φ-unfolding structures can be consistent with the same complex and φ-values. Several of these dynamic possibilities may be worth exploring in future IIT research, as part of exploring how those dynamic transitions would be experienced (or recalled) by a given phenomenal self.

For this paper, the only dynamic system aspect we need to analyse is the shortest path (p) between the sets of units constituting the main complex (c_main_) at two adjacent points in time (t_1_ & t_2_; c_t1main_ & c_t2main_):p(c_t1main_, c_t2main_) = argmin_u,v_{d(u,v) ∣ u ∈ c_t1main_, v ∈ c_t2main_}; p ∈ ℤ_≥0_
where d(u,v) is the shortest path length between units u and v (i.e., the min. number of arrows between u and v in the grammar of Figure 1)

Three mutually exclusive, collectively exhaustive possibilities capture the option space of interest for this paper:a path distance of 0 (i.e., at least one unit is the same across the main complexes at time 1 and time 2)a path distance of 1 (i.e., the nearest unit in the main complex at time 1 is a single arrow edge away from the nearest unit in the main complex at time 2, having checked all possible paths)a path distance of 2 (i.e., the two main complexes do not share any units or adjacent units).

Strictly speaking, allowing the second possibility excludes trivial graphs of 1 unit and allowing the third further excludes tiny graphs with 2–3 units, wholly connected graphs of any number of units (each unit connects to every other unit), and graphs so richly connected that no path distances of 2+ exist. Given our focus on human phenomenology and IIT’s principal empirical focus on neurons in the human brain, we do not consider these exceptions to weaken our argument. In other words, given current understanding of IIT and the human brain, we consider all three of these possibilities to be plausible in a human setting. The possibility that the relevant causal graphs for phenomenal binding always overlap is discussed in Section 5. The possibility that the relevant graphs for phenomena binding are trivially small is discussed further in Section 4.3.

Figure 3A–C provides an example of each of these three possibilities, working with the same abstract graph as Figure 2.

If we imagine such transitions from time-step 1 to 2 as reflecting a change in the human brain, it is reasonable to ask what is happening to our experience in each of the three possibilities, given that IIT traditionally identifies the awake and self-reporting human self with the main complex (e.g., [49] p. 17, [53]), although the theory and mathematics remain self-consistent if it does not.

The idea that conscious experience persists over non-zero temporal periods is intuitive, because a ‘zero’ duration cannot contain anything at all (although contested, e.g., discussion in [54]). Equally, it is intuitive that the same self (for some sense of ‘same’) experiences dynamically changing content over time. The self that is reading this paragraph of the paper retains a link to the historic self that read the prior paragraph. While these dynamic conscious phenomena may not be what they first appear, explaining their appearances is a reasonable first step. For instance, leading IIT theorists have developed an account of the phenomenology of time, such that its directedness is the result of asymmetries in the causal structure of a 1D directed grid [49]. Their focus is on how the flowing experience within an extended present can be created within a static causal grid, i.e., phenomenal time as a structure not a process. These efforts are complementary to this paper, which focuses instead on the succession of entities as a structure evolves over clock-time.

In the context of these dynamic conscious phenomena, path distances of zero can have a straightforward interpretation. My conscious experience remains in the same main complex in both time steps. The content of my experience might change, as well as the volume of that conscious experience. For instance, example A in Figure 3 sees an expansion from units JKPQ to JKPQL. Depending on what is happening to the φ measure of integrated information and the Φ-structure implications, this change might correspond to a sense of ‘expanded awareness’. We know that our dynamic self can experience phenomenal changes in this type, so it is tantalising to see a potential physical/structural correlate emerge from IIT’s theoretical infrastructure. Even a more substantial change might have an intuitive correspondence. A shift from JKPQ to JIHS still contains an overlapping unit and perhaps reflects some of the sharp shifts in attention or self-boundary modelling that we can experience.

Path distances of one are a little less intuitive, but we can still imagine a dynamically persistent self that moves from one location to an adjacent one. In IIT, the self in a given moment is a physical entity and there is nothing unusual in the boundaries of physical entities shifting over time as the entity moves through space. Possible parallels to support the intuition include the spreading of a fire to a new location while it burns out in the original location or a macrophage moving to engulf a bacterium (phagocytosis).

### 3.3. An Aside on the Temporal Dynamics Within Complexes

Before we turn to path distances of two or more, the discussion above provokes an aside that may be of interest to IIT theorists (please skip to 3.4 to continue the core argument directly). Where any adjustment of physical boundaries takes place, it is reasonable to ask a future version of IIT to specify the temporal dynamics by which the adjustment propagates through the cause-effect structure and resulting distribution of complexes.

In its current static presentation, a single time-step contains a potentially complex conscious experience for a given entity, as defined by the Φ-unfolding of its complex. This Φ-unfolding depends on hypothetical causal transitions in the TPM. It does not depend on any of these causal transitions actually happening over a period of time. The experience is instead static and distributed over the quantised time step for the entity that exists corresponding to that time step. Note that time is discrete in IIT, as with space, and the relevant temporal grain at which the entity exists is identified as the φ-maximising value across all possible temporal grain options.

As we move into a second time step of the φ-maximising temporal grain, the physical substrate incorporated in the successor to the main complex might span a significantly different spatial distance than the original. However, the new Φ-structure exists instantaneously across that new physical space in the new time step—at least the way the current calculation process is described.

If there is any dynamic continuity of entity self-hood between two time-steps, such instantaneity across physical substrates raises questions given our current understanding of physical constraints, notably limits against faster than light information transfer due to the theory of relativity. If we imagine the successor entity encompassing a space that is 10 cm extended from its predecessor (a plausible distance in the human brain, given IIT’s interest in the posterior zone [55]), it would take around 0.33 nanoseconds for light to transit that in a vacuum (and significantly slower in the brain substrate). While much shorter than the likely temporal grain in any φ-maximising analysis [49], this duration remains far longer than any hypothetical base/micro units of time.

As the physical space encompassed in the main complex expands, perhaps in response to a state change in a given unit, the physical propagation of that state change will take place within the cosmic speed limit and span multiple base/micro time units. Only when the propagation is complete would the new φ-maximising complexes be in place. Until the propagation is complete across the system, it is not obvious what the distribution of complexes should look like. We have not yet completed a full time-step of the previously φ-maximising temporal grain, so cannot impose the complexes that emerge at that time step and exclude any constituent time units: the future has not happened yet and that full time step does not yet exist (maybe it never will).

What happens during this period of propagation? Are there millions of tiny conscious moments across microscopic selves in between the establishment of Φ-complexes across the adjacent but relatively sluggish time-steps at the φ-maximising grain? What temporal gaps or spatial grey areas in experience propagation might need to be addressed? These questions are particularly poignant with entity continuity between time-steps but need to be addressed even within an individual time-step for an entity corresponding only to that time-step.

Given that φ depends not just on actual causal transitions but also on hypothetical causal transitions, we cannot lean simply on the slowness of physical relationships in the solution. Possibility alone might obey logical rather than physical limits. A slow φ-maximising temporal grain provides some resolution of functionally (but not physically) instant integration within an individual time-step but is not obviously neat in its resolution of multiple time-steps.

Certain positions in the metaphysics of time might be available to tackle this challenge. In a block universe with a fixed future, perhaps the φ-maximising temporal grain can be simultaneously imposed across the whole system. However, if the future is fixed and uncertainty exists only as an epistemological construct for certain observers, it is unclear that IIT can build ontology-driving mathematics on hypothetical causal relationships based on the uncertainty of causes/effects given a particular state. Moreover, even if there are no interim moments to be addressed, the propagation of new states still needs to be constructed in a way that aligns with cosmic speed limits.

We suspect IIT theorists will be able to identify a mathematical fix for this situation and describe the corresponding dynamics of integration. Nonetheless, there are multiple possible fixes, motivating the value of theoretical attention and potentially shedding light on the broader dynamism that is the focus of this paper.

In the same spirit of fleshing out dynamic continuity and given IIT’s recognition that its approach to tie-breaking remains open to evaluation and perhaps to further development [2] (p. 40), a future version of IIT might want to explore what happens when a main complex effectively splits into two in the following time-step. Discrete topological bifurcations of the main complex could also be constructed in continuous time. In such circumstances, what do we experience? Which successor complex does the self follow? The ideas in this paper provide a few possible routes to follow, but others may be preferable, such as moving down to the ‘next big thing’ when resolving which of multiple, equally valid, larger complexes should be chosen [56] (p. 11). It is for IIT theorists to motivate one particular route in a clarification of IIT v4.0 or an evolved IIT v5.0.

### 3.4. The Dilemma Under Path Distances of 2+

For this paper’s core argument, however, we focus on the most challenging example in Figure 3, acknowledging that certain resolutions might also address the points in Section 3.3. The most challenging example is what happens to conscious experience when we have two main complexes that are temporally adjacent but not spatially adjacent: the ‘dynamic entity evolution’ problem. And, where this happens in a human brain, what happens to our dynamic sense of self? The is example C in Figure 3, i.e., a path distance of two or more between temporally adjacent main complexes. Unlike the interpretative flexibility open to examples A and B as described above, this situation can be presented as a dilemma of two horns, where each horn has potentially unpalatable consequences.

In horn one of the dilemma, our original self remains in the area contiguous with the original complex. A new, separate self emerges in the new complex. Note that IIT has always accepted the presence of multiple conscious entities (i.e., non-zero-φ complexes) within one biological brain—and indeed within much simpler physical systems, so this alone is no more unpalatable than already confirmed as part of IIT’s architecture. However, in this case, the new entity is larger in terms of integrated information and volume of consciousness. If it were to nest our original self, our self would be eliminated by the principle of exclusion, leaving only the new entity (this paper is not focusing on nesting; the point is merely to emphasise the significance of the new entity). In other words, we must accept that our human consciousness is not always the largest consciousness in a biological brain.

At face value, horn one may seem like a reasonable bullet to bite—perhaps it also provides an account for topics in psychology such as tulpas, IFS (Internal Family Systems), or split personalities. But there is more to this bullet than a potential disquieting humility, there is also an experimental cost. One reasonable short-cut in IIT’s neuroscientific experimentation is to assume that the largest φ-structure in the brain will correspond to the subject’s seat of consciousness, opening up avenues for mapping that specific structure to the contents of conscious experience in that moment and even for manipulating that structure and reinforcing the theory’s validity through self-report.

Looking for the largest φ-structures in the brain helps to motivate heuristics for ruling out parts of the brain, which is currently essential given the computational intractability of IIT’s mathematics when applied to large parts of the brain. However, under horn one of this dilemma, it might be that the conscious entity who provides the reports in the experiment is identical with the second largest φ structure in the brain—or some even smaller complex. Examining all these smaller complexes is a more challenging empirical and computational task. Nonetheless, it is important to emphasise that this implication or its experimental cost do not contradict anything in IIT’s existing theory. IIT claims that there is a complex whose cause-effect structure corresponds one-on-one to the properties of our experience, not that this structure is any particular size in the brain (or even necessarily contained within a single brain) (we thank a reviewer for this emphasis). The experimental task is to find that complex and remains possible in principle even under horn one of the dilemma.

In horn two, the self moves to the new, larger φ-complex. However, by definition, this must be a non-local movement. There is no time and no physical rationale for it to pass through the intervening physical space. What is this self if it is subject to the laws of physics as we understand them and if it can jump to a new location without transiting the intervening points? One possibility is that the self is something non-physical so it is not bound by the usual physical laws that operate at the human scale. However, IIT’s existing metaphysics of intrinsic and extrinsic existence will need adjusting to incorporate this ‘non-physical, non-local’ nature and explain how these different types of entity/existence interact, taking care to avoid the challenges that can result from dualist metaphysics (see overview in [57]).

A second possibility is that there is something fundamentally quantum mechanical about the self, potentially enabling theorists to draw on insights about non-locality in quantum mechanics (although the ontology of quantum physics is far from settled; there remain interpretations that reject non-locality). However, in our current understanding of physics, if the self is to have uniquely quantum mechanical properties, it must be extremely small or be protected by some remarkable physical structure that maintains the relevant quantum properties at meso-scale.

Perhaps IIT can be reformulated at a micro-scale—its mathematics are not scale dependent—but this would constitute a major shift away from the leading IIT account in which neurons form the base units of human consciousness. Similarly, if quantum mechanical features are necessary conditions for self-definition, this would also be a major extension of IIT. However, it is worth flagging that IIT researchers have already begun exploring applications of IIT concepts in the context of quantum phenomena [58,59].

A third possibility is that the seemingly non-local transition between one time-step and the next is actually an artefact of temporal macro-graining [37] (We thank a reviewer for surfacing this possibility explicitly). When examined at the micro-scale, there would in fact be a series of micro-transitions, each of which involves only local dynamics, with the self moving locally each time.

If coarse-grained time-steps are allowed some degree of fuzziness (in some elaboration of IIT v.4.0), there might already be enough space to permit this kind of solution. Alternatively, in a canonical interpretation of IIT, once the final micro time-step is in place, the calculations adjust such that multiple time-steps are coarse-grained together into multi-time-step, multi-unit complexes, which so happen to manifest the new target complex which appears, in hindsight, to be spatially non-contiguous with its predecessor. Viewed from the present, those constituent micro-time-steps no longer exist intrinsically—they have been subsumed into the new macro-time-step complex via the same principles discussed in Section 2.5. Nonetheless, as the processes unfolded in real-time, those micro-time-steps did exist and did permit the necessary local transitions.

For IIT to pursue this third possibility, further detail is needed on the dynamic transitions that constitute coarse-graining, building on [37], while addressing this distinction between complexes ‘viewed in hindsight’ vs. as micro time-steps unfold and specifying a solution for the related issues described in Section 3.3.

## 4. Dissolving the Dilemma

Section 4 describes three broad approaches for dissolving the dilemma, without having to accept any of the (potentially) disquieting implications of the two horns described in Section 3. The first denies the validity of introspective evidence of short-term dynamic persistence, perhaps embracing empty individualism instead. The second dissolves the dilemma by moving away from the metaphysical solution by which IIT addresses the binding problem. The nature of the second approach depends on the new metaphysics that replaces it; three examples are briefly provided, recognising that other metaphysics could also be explored. The third approach denies the need for phenomenal binding in the first place.

### 4.1. Empty Individualism to Dissolve the Dilemma

Under empty individualism, our experience of a dynamically persistent self is an illusion (e.g., discussion in [3]). Instead, we exist only as individual, separate, and single time-step experiences. A new question arises as to how to explain our illusion of dynamic persistence, but options are available. For instance, provided the single time-step has non-zero temporal duration, a pseudo time arrow can emerge from repeated access to an immediate memory module (see Section 4.3.3 of [60]). There is also a broader literature on philosophy of time and the phenomenology of time which might provide alternative routes to a solution (overview in [61]).

What would this mean for IIT in the dilemma? The change in ‘main complex’ in Figure 3C still happens, but there is nothing actually ‘transferring’ between them, so no need to identify dynamics that avoid (or explain) faster than light information transfer. Each time-step brings into being a new zoo of single time-step selves: one per complex. Even if there are larger experiences from time to time in the human brain than the one corresponding to us, they are not linked to the same memory module and so would not be identifiable as part of the illusory dynamism in our self-hood.

The challenge to IIT’s approach to neuroscientific experimentation from horn one of the dilemma remains in place: that our subjectively reporting self may not always be the highest φ complex we can identify in the human brain, although it must be somewhere in the system and can be identified experimentally, at least in principle. However, depending on how illusory dynamism is implemented, we have clear direction on where else to look: the φ-defined main complexes overlapping with the immediate memory module or systems transferring information with it.

### 4.2. A Different Metaphysics to Dissolve the Dilemma

The second way to dissolve the dilemma is to move away from the metaphysical solution by which IIT addresses the binding problem. The implications of this approach depend on the new metaphysics that replaces it. We briefly describe three possibilities here in order to highlight that neither provides an easy route to a solution. However, full discussion of these and other metaphysical approaches would need to be addressed in separate papers.

Possibility one is to reframe the role of complexes. If complexes are no longer the material ontological unit driving causality, there is no need for non-locality because nothing is actually moving. In this possibility, micro units continue to exist intrinsically even when part of a complex; complexes are ‘analytical’ rather than ‘ontological’ phenomena.

Unfortunately, if micro units continue to exist and have all their normal causal power in their interactions even where they form part of a complex, then the local causal relationships embedded in IIT’s causal graphs (i.e., direct unit-to-unit interactions only) are sufficient to explain all causal behaviour at the complex level. In other words, whether a particular complex has highest integrated information becomes a purely denotative feature that does not change what the system does overall. Such denotative features can be analytically useful and exist in the eye of the beholder (or ‘extrinsically’ in IIT terms), but they do not affect actual causality by definition. The mathematics of IIT may remain valuable to scientists under this possibility (e.g., a ‘weak IIT’ interpretation, see [4]), but IIT has lost its solution to the phenomenal binding problem. There is nowhere in the new metaphysics where multiple units of information come together into a single point of view with intrinsic existence (only micro units have fundamental ontological existence).

Possibility two maintains the ontological persistence of micro units, while interpreting complexes as patterns of those units. Complexes do not have a fundamental existence as objects, but they do exist as patterns. This would be a third type of existence to add into IIT, in addition to ‘intrinsic’ and ‘extrinsic’ existence. From this perspective, consciousness itself is also understood as a pattern, rather than an entity.

This second metaphysics denies IIT’s intuition of consciousness as intrinsic, so may prove a step too far for its advocates. However, pattern or process perspectives on experience have proved intuitively convincing to influential thinkers in the field (e.g., Dennett, Hofstadter, Minsky etc.).

Such a theory needs to begin by explaining this third type of existence in a principled, mathematical formalism, relating it to IIT’s existing postulates. There are also potentially challenging metaphysical questions to answer about what ‘patterns’ even are, outside of the eye of a beholder or an entity who can see the pattern ‘all at once, i.e., without presupposing a solution to the phenomenal binding problem.

Possibility three dissolves the problem by introducing a different notion of ‘self’ (we thank a reviewer for introducing this possibility). Rather than identifying each individual self with a single complex, we allow multiple complexes to each pick out the same self. A single self might be associated with multiple complexes within time-steps as well as across them. Such a self would by definition have a distributed extent across multiple locations and times, removing the need for non-local transitions from one complex to another. A purely informational self might collapse into a memory solution (see Section 4.1), but it is unclear how such a self can play the role required as a seat of phenomenal awareness that removes the need for non-local movements in awareness. Instead, the ‘self’ needs to be a phenomenal self in the sense of being a locus of our conscious awareness.

Work would be needed to motivate these ‘phenomenal selves’ theoretically and specify their mathematics as part of an expanded formalism. For instance, these selves might operate in a third level of an existence hierarchy, similar to how multi-unit complexes can be interpreted as existing in a second level compared perhaps to their constituent (and wholly subsumed) micro units or to single-unit complexes (which have their own intrinsic existence at the entity-level).

Unfortunately, even if the issues above can be addressed, all three of these metaphysical positions share a worrying bullet to bite. Causality has been disconnected from the level at which we traditionally associate our conscious self. In possibilities one and two, causality remains fully explained via the micro units that exist fundamentally and persist no matter what complexes they form part of. In possibility three, causality remains fully explained via lower levels of existence, with none left for the phenomenal self. In other words, these metaphysical adjustments denude macro-scale conscious entities of the causal power that is central to IIT’s current axioms—and indeed corresponds with our felt experience of agency.

Epiphenomenality of human-style consciousness may be too steep a price to pay, especially given IIT’s emphasis that it is one of the few neuroscientifically grounded theories such that “we have true free will—that we have true alternatives, make true decisions, and truly cause. Because only what truly exists (intrinsically, for itself) can truly cause, we, rather than our neurons, cause our willed actions and are responsible for their consequences” [2] (p. 40).

The implication of epiphenomenality (at the level of emergent structures) that follows from wholly defined local causal interactions is a well-studied topic (overview in [62]). Various escape routes have been devised, although all are contested. IIT could explore such routes, but it seems hard to find a solution while maintaining the heart of what makes today’s IIT a compelling theory for many. For instance, IIT might want to introduce a form of strong emergence to provide novel causality at the complex or self level, but this would entail a major departure from the unit-level TPM-led approach of IIT’s current approach to causality and require grappling with the well-known philosophical difficulties of strong emergence (overview in [63]).

### 4.3. Deny the Need for Phenomenal Binding

A further option is to dissolve the phenomenal binding problem—or at least reject its force. If our actual experience in any given moment is not actually that complex, perhaps it can be adequately captured by a single micro unit or at least by a certain type of micro unit that has higher informational complexity than IIT’s assumption of two states. In other words, no phenomenal binding *between* units is ever needed. Perhaps we can incorporate that unit’s *direct* causal relationships with other units to increase complexity somewhat. For instance, while an individual unit remains the locus of awareness, the content of that awareness might incorporate some set of that unit’s direct ‘single edge’ causal relationships, albeit not the units on the other end of those relationships (which simply reintroduces the need for a phenomenal binding mechanism).

This approach may be worth pursuing, but needs to contend with the prevailing neuroscientific view that our conscious experience seems to reflect information in multiple parts of the brain and there does not appear to be one specific micro unit where all consciously experienced information collects (e.g., [9]). It also needs to contend with the prevailing view in particle-based physics that whatever the ultimate units of reality turn out to be, they are likely to have only modest informational complexity. The journey in particle-based ontologies is towards units in the Standard Model of Particle Physics with relatively few degrees of freedom and a modest set of interactions. Such simple units seem to conflict with our felt experience of complexity in individual moments but might be addressable if that felt experience is only illusorily complex.

Could the complexity in our felt experience be an illusion? After all, some have argued that the informational capacity limit on our thoughts is much narrower than we instinctively feel (e.g., [64,65]). A possible parallel might be found in the visual system. Our visual field feels complex, but saccade models can be interpreted to suggest we only see tiny informationally simple parts at a time. The full apparent complexity is not actually observed all at once, rather only when we turn attention to small subsets of that complexity at a time. The apparent complexity is a phenomenal illusion, in contradiction to the motivation in this paper.

## 5. Conclusions and Future Research

This paper explores the possible theoretical interpretations of main complexes being in different parts of the brain over time, e.g., as inputs vary, system states vary, or neuronal cells/connections are reconfigured. We identify several options, each of which has potentially unpalatable consequences for IIT advocates. Table 1 summarises the discussion into five main options and the primary consequences of potential concern.

Our presentation of these consequences is primarily intended for theoretical clarity, rather than as direct criticism. IIT advocates have an admirable track record of embracing theory implications that strike some observers as counterintuitive, such as the presence of some minimal phenomenal experience in an electron or simple logic gate infrastructures (e.g., embraced in [53,66]), the sensitivity of consciousness to tiny system changes (criticised by [67]; embraced in [2], p. 39), tolerance of zombie systems that have identical behaviour but none of the experience of conscious systems (criticised in [68] s4.2, embraced in [69] Figure 21, [70]), and the dependence of conscious experience on neurons that never fire and will never fire (but are capable of firing in theory) (embraced in [2] p. 40).

Nonetheless, some researchers might move away from IIT as a result of these consequences or others, such as concerns over the functional irrelevancy of experience ([68] s2.2), a possible paradox between IIT’s irrefutable introspection and uncertainty about consciousness [2] (s4.3), noting the possibility of a metacognitive solution [2] (s4.4), or other metaphysical issues, such as dealing with the phenomenal simultaneity of consciousness with physical reality [71].

A key insight from our paper is that the exclusion postulate cannot be easily removed from IIT. For instance, ref. [72] suggests that IIT remove this postulate, in order to have more flexibility in considering how individual consciousnesses (such as humans) might be nested within larger structures that might meet the standards for consciousness equally well (such as a large economy) [72]. However, our analysis links the exclusion postulate and its corresponding metaphysics to IIT’s solution to the phenomenal binding problem. Without exclusion, IIT cannot easily explain the complexity of consciousness that is common to the human experience—at least not without embracing epiphenomenality. These are much steeper prices to pay than those Schwitzgebel describes. As with all metaphysics, alternative accounts exist—including those summarised in Table 1—but none is straightforward.

In the absence of major metaphysical readjustments, IIT theorists may be particularly drawn to options 1–3. They may also wish to pursue an empirical research avenue that could greatly mitigate the potential costs associated with option 1. It is possible that healthy human brains are structured such that all main complexes in all historically experienced pathways have always been overlapping with each other. In other words, while we can imagine (and design) a system with the problematic main complex dynamics described, such dynamics never actually occur in the human experience. As a result, it is not surprising that we lack an intuition for how such dynamics might be perceived. It may still be theoretically productive to analyse these dynamics, but the ‘potential consequences’ listed in the table do not apply in practice in the human setting and should not be considered a barrier to adopting option 1.

In options 1 and 2, there are also questions over avoiding faster-than-light information transfer in experience evolution between time-steps, such as occur even for locally overlapping or locally adjacent shifts in main complex over time (Section 3.3). However, a version of those questions applies to any dynamic analysis of IIT, regardless of assumptions on entity continuity and the options in Table 1. The unfolding of Φ-folds from the start to the end of a single time-step raises propagation questions, whenever the φ-maximising spatio-temporal grain is larger than the base unit, including IIT’s neuronal models of human consciousness. It should be possible to develop information propagation dynamics into IIT’s theory to prevent this problem, but it is not currently available. During these propagation dynamics, it is plausible that different φ values and distributions of complexes might be temporarily present—these circumstances also need exploring. Addressing this question is a useful future research avenue, to help IIT move from a primarily static presentation to a dynamic theory.

Our view, reinforced by conversations at conferences in 2024, is that different IIT theorists or observers may favour different positions out of those outlined in this paper and summarised in Table 1. Even if the correct option seems obvious to some researchers, we hope that making such a position explicit, understanding its implications, connecting the conceptual position with IIT’s mathematical formalism, and conducting supporting research will help IIT continue its development. IIT is already on version 4 of its development and is to be respected for remaining open to further refinements.

## Figures and Tables

**Figure 1 entropy-27-00338-f001:**
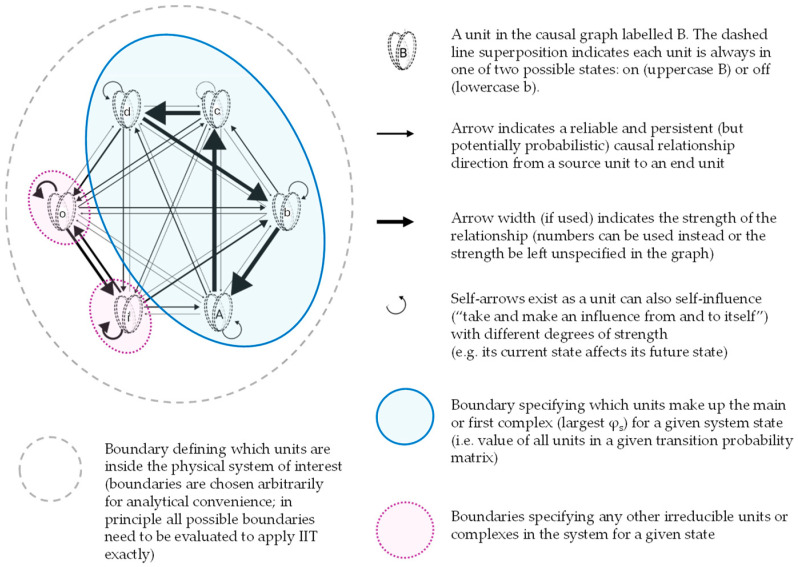
Substrate graph with annotated grammar. Source: Adapted from Abcdio training example in [52].

**Figure 2 entropy-27-00338-f002:**
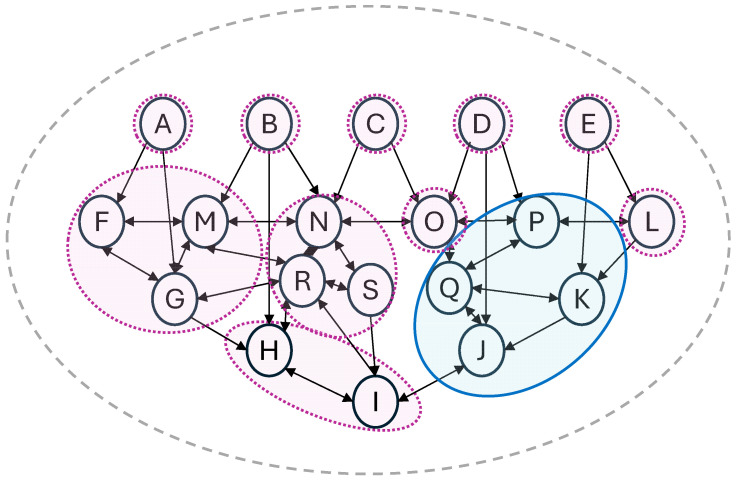
Illustrative substrate graph with diverse complexes. See Figure 1 for legend. Note: Self-arrows present for all units; suppressed for legibility.

**Figure 3 entropy-27-00338-f003:**
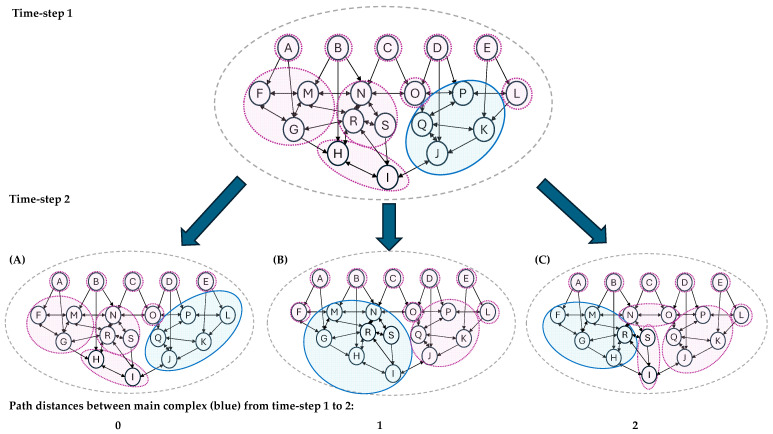
Illustration of three path distance possibilities across time for main complexes. (**A**) illustrates a path distance transition from time-step 1 to time-step 2 of 0. (**B**) illustrates a path distance transition of 1. (**C**) illustrates a path distance transition of 2.

**Table 1 entropy-27-00338-t001:** Options for resolving the dynamic entity evolution problem and consequences.

How to Interpret the Main Complex Evolving Non-Locally over Time in a Given System?	Potential Consequences for Theorists to Address or Dissolve
1. The new main complex is a new experiencing entity, separate to any successor to the previous main complex	– The subjective self we identify with in the human brain is not always the highest φ complex in our brain – IIT’s experimental paradigm cannot rely on the convenient heuristic of looking for the highest φ part of the brain (but IIT’s theory remains internally consistent despite this)
2. The experiencing self identified in the previous main complex seemingly moves non-locally to the new main complex	The nature of this apparent non-local movement needs to be specified in a way consistent with known or hypothesised physical laws. – e.g., 1: If the self is non-physical, IIT’s metaphysics needs to be adjusted to explain this non-physical nature in the context of IIT’s existing theory – e.g., 2: If the (human-relevant) self exists only at a scale where quantum mechanical non-locality operates, this needs to be incorporated into IIT’s framework – e.g., 3: If macro-time-steps are constructed from micro-time-steps such that complex transitions in micro-time-steps always maintain local continuity, the micro-macro mathematics needs to be specified accordingly
3. Selves do not persist dynamically over time (e.g., empty individualism), so there is no ‘locus of awareness’ to transfer	– The introspected experience of dynamic persistence needs to be accounted for as an illusion – IIT’s experimental paradigm faces the same difficulties as (1) but might have an alternative convenient heuristic to apply (e.g., main complexes overlapping or connected to an immediate memory module)
4. Adjust the metaphysics, e.g., interpret complexes as analytical structures, as patterns, or as connected to a separate multi-complex ‘self’	– Major departure from IIT’s existing metaphysics, e.g., of complexes as ontological structures, of consciousness as intrinsic and fundamental, or of complexes as the ‘top layer’ of the ontological hierarchy—details/implications need working through – Epiphenomenality of human-style consciousness needs to be accepted or otherwise addressed, perhaps via the introduction of strong emergence
5. Deny the need for phenomenal binding because adequate complexity for consciousness can be present in a single causal unit	– Need to explain the mechanisms that make introspected experience appear illusorily complex – Need to reconcile with the prevailing views in neuroscience of distributed conscious information and with prevailing views in particle physics of base units with limited degrees of freedom

## Data Availability

The data are contained within the article.

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
