# Peer review of "Integrated Information Theory and the Phenomenal Binding Problem: Challenges and Solutions in a Dynamic Framework"

_entropy, 2025, doi:10.3390/e27040338_

Round 1

Reviewer 1 Report

Comments and Suggestions for Authors

The article offers a novel and useful analysis of IIT's implications with respect to what the authors call the dynamic entity evolution problem. The authors start by outlining how the current version of IIT (4.0) provides a solution to the phenomenal binding problem, they then consider a system in which the first complex changes across time-steps, and consider various answers to the question about continuity of complexes across time.

Full disclosure, I am one of the developers of IIT. I find the authors' attempt to provide constructive criticism to further improve IIT very useful, and I will use my knowledge of IIT mostly to point out minor inaccuracies in the presentation of the theory and to suggest further relevant references.

The article is well written, the authors correctly characterize IIT's core claims, and offer a useful investigation of alternative ways to solve the dilemma they propose. For this reason I recommend publication with minor revisions.

Main comments:
Overall, the article does a good job in separating claims that can be directly reconducted to the current version of IIT, from more speculative explorations of IIT's implications or of possible alternative claims IIT or other theories could make. I have some comments that I hope will help the authors improve the paper.

First, the paper seems to imply that, for IIT, the main complex corresponding to our experience should also be the first complex, i.e. the max phi_s complex in the brain. This is a reasonable assumption, but according to IIT it is not necessary. For IIT to be correct, the only requirement is that each experience is fully accounted for by "a" complex unfolded (which is of course a maximum), not necessarily the first complex (the "maximum among maxima"). In light of this, the claim at page 18 L643-645 that "The heart of IIT’s approach to neuroscientific experimentation is based on the idea that finding the largest Ï•-structure in the brain will allow us to identify the subject’s seat of consciousness" is not fully correct. In fact, why should we limit the search within the brain and not identify our experience with the first complex in the body, or even outside of it? The point of IIT is that, as long as we find a complex whose cause-effect structure corresponds one-to-one to the properties of our experience (both essential and accidental), and the complex is a global maximum (i.e. not overlapping with higher phi_s complexes), then we have identified the substrate of that specific experience.
The authors consider this options and in fact proceed to say that (L648-650) "it might be that the conscious entity who provides the reports in the experiment is identical with the second largest Ï• structure in the brain – or some even smaller complex. We no longer know what to look for in our lab experiments." According to IIT, we do know what to look for: a complex that accounts for all properties of the specific experience under investigation, which could indeed not be the first. Thus, this option does not seem as problematic per se, and the claim that it comes with an an experimental cost (also repeated in the context of section 4.1) does not seem cogent. If the authors consider the criticism still valid, I would recommend to clarify in order to make the point stronger.

A second point is related to the authors' claim that, in IIT, existence means something different from the common intuition (L391). The authors are correct, for IIT, "intrinsic existence" or "true existence" only applies to conscious entities. Nonetheless, IIT grants that extrinsic entities like tables and chairs exist in an extrinsic sense: they exist only for us (or other intrinsic entities), while they themselves have only extrinsic existence. To clarify this point further the authors can consult Tononi et al. 2022 https://arxiv.org/abs/2206.02069 Moreover, Mayner et al. 2024 https://arxiv.org/abs/2412.21111 provides an account of why and how we perceive these extrinsic entities, doing justice to our pre-theoretical intuition.

A third point concerns single unit complexes. It follows from IIT's formalism that, in order to exist, complexes must be constituted of "intrinsic units" which themselves comply with the postulates of IIT, which can be constituted of micro units (not further constituted of more smaller units). In the limit, micro units must also comply with the postulates in order to exist, a point that the authors discuss in footnote 4. The paper seems to suggest otherwise. The authors can consult Marshall et al. 2024 https://www.biorxiv.org/content/10.1101/2024.04.12.589163v2 for clarifications on these points.

Minor points:
L90-91 "maximally irreducible conceptual structures (‘complexes’) define existence for experiencing subjects"
The authors mostly use IIT 4.0 terminology. "Conceptual structure" is an IIT 3.0 term, I recommend to substitute with "Φ-structure". Moreover, technically a Φ-structure is the cause-effect power specified by (or unfolded from) a complex, and is not identical to the complex itself.

L140 I believe Roskies would use she/her pronoun

L275 "complexes can be understood as sets of atomic units". "Intrinsic units" would be more appropriate here in light of Marshall et al. 2024 (see above)

L309 "Individual units of (potential) information". I see why the authors use this language. In IIT however it would be more correct to speak about units of integrated information or cause-effect power, or to be more technical about "causal distinctions", which together with causal relations are the components of a Φ-structure.

Footnote 13: IIT allows for single unit complexes (and indeed requires that monads are just that)

L844 Concerning zombie systems, the authors can also consult Grasso et al. 2021 https://doi.org/10.1093/nc/niab022 and Findlay et al. 2024 https://arxiv.org/abs/2412.04571

Reviewer 2 Report

Comments and Suggestions for Authors

This paper is creative and addresses a significant problem in consciousness studies, related to the (phenomenal) binding problem, the problem of continuity of the self between instances of experience, the way IIT addresses these issues, and finally, a “roadmap” for moving forward.

If I understood the problem correctly, it goes as follows: According to IIT, the locus of experience ("what truly exists") is picked out by a “complex”. However, across time, this complex can change (or it can stay the same but the related experiential content might change). How to have a notion of self that is (or is not) permanent across time in this setup (the “dynamic entity problem”, DEP, is specifically raised in the scenario where two complexes are temporally adjacent without being spatially adjacent)? (At least, phenomenologically, this should be the case, so at least this permanence should figure among some experiential structures).

If one chooses to say that “we” always remain “in” the same part of the substrate, “we” might dissociate from the complex (i.e. there are other consciousnesses present in the brain than ours that are much larger in phi). The alternative is to say that “we” shift our “location” in the brain according to where the main complex is. Authors call this a “non-local” movement, but I remain unconvinced. The underlying information transfer in the brain is most probably still local (the “timestep” most likely being an approximation to some local dynamics, also state-changes do not happen instantaneously on the level of the brain), unless one explicitly invokes quantum phenomena 9or a quantum extension of IIT, see

  • Albantakis et al. Computing the Integrated Information of a Quantum Mechanism, Entropy, 2023.
  • Kleiner & Tull. The Mathematical Structure of IIT, Frontiers in Appl. Math. Stat. (2022)

Three possible dissolutions of the dilemma are offered: (i) give up on the notion of permanent self (“empty individualism”); one ends up with snaphots (me1, me2, m3) over time, (ii) give up the metaphysics of IIT in favor of, for example, standard non-reductive physicalism (this has the nasty side-effect that much of IIT becomes invalid, in particular its solution to the problem of unity), or (iii) get rid of phenomenal binding (only phenomenal units are single particles (again, this would imho undermine IIT). To me, therefore, it seems that only option (i) is feasible for the integrated information theorist. (but others might disagree)

Throughout the paper, an important experimental consequence is highlighted for IIT, namely on some of these options: experimentally testing “whether I am still in there” becomes impossible. There is almost always someone “in there” but it is almost surely not me. This is a problem if one wants to use IIT for deciding whether to turn off the instruments.

That all said, some changes are necessary to further the manuscript's impact:

  1. It is important to note that, at least in previous iterations, IIT was often presented as a “snapshot” model of (temporal) consciousness. By default, such approaches have difficulties to amount for any continuity of the self. This is worsened by the fact that in IIT, the minimal subject is picked out as complex that might, as the authors correctly note, shift in time across the brain. However, it leaves open the question how the notion of a complex is tied to the “thisness” of the self (which seems basically to be the question of the authors). Couldn't it also be the case that all complexes (at least those that could potentially figure in the same substrate) are picking out the same “self”. Would this be another (metaphysical) option for IIT?
  2. Throughout the paper, the authors give a sense in which existence is defined by “taking and making a difference”. I am not sure as to whether this is really a “definition” of existence (a la Eleatic principle), but rather an operationalization of existence in physical terms (as IIT-people would probably want to put it now).
  3. On the Robustness of the BP to the notion of phenomenal unity (l 221 ff.) This is a good point. However, if one were to give up the notion of unity, what would that entail for the notion of self? Without such a notion, much of the problem is void, no? So, what is, specifically, the relation between the DEP and binding? Isn’t it a problem related to unity (at least for IIT)?
  4. Doesn’t IIT’s integration axiom speak to unity rather than binding? (One argument for its irrefutability is that one could not conceive of a disunified consciousness because, as you put it, a “fragmentation is itself part of the unified experience”.) It seems then that, following Dennett et al., if one gives up unity, one gives up a central ingredient of IIT. Indeed, you seem to be correct that the idea of unity could be refuted while still holding up the phenomenal binding problem. What seems less clear is that one can buy into IIT if one refutes the idea of the unity of consciousness. (and hence the problem you mention would not arise, because IIT is anyway not a viable option)
  5. The paper devotes a lot of space to footnotes. This is typical of humanities papers but might not be suitable for Entropy. I suggest either removing the footnotes (in particular if they are a little digressive) or else incorporating as much as possible into the main text (the typical Entropy reader will not read the footnotes, so all the essential information in them will be lost).
  6. L318 “classical mechanics”; this sounds a bit as if IIT would not conflict with intuitions from quantum mechanics, which is however doubtful.
  7. L331 This is essentially the point of Hoffman’s Interface Theory:
  • Hoffman et al. The Interface Theory of Perception, Psychonomic Bulleting & Review (2015)
  • Prakash et al. Fact, Fiction, and Fitness, Entropy (2020)
  1. The (somewhat idiosyncratic) discussion of mereology in Sec. 2.6 is a bit digressive. Relevant points (l. 359ff.) could be incorporated into the previous section 2.5.
  2. l. 423 This sounds a bit misleading, as if there were just one “true” complex and many single-unit ones. But this is probably not what you mean? What does Fig. 1 or 2 actually add? One could probably condense this. It appears that everything that comes before l. 440 is not really needed for your discussion.
  3. L 655 Saying that the self is non-physical does not imply (substance?) dualism. There are many other options here. IIT seems to endorse one of them (though it is not quite clear which one).
  4. Subsection 4.2 seems to tackle a huge question (developing a new metaphysics for an IIT-inspired but fundamentally non-IIT) within a few paragraphs. I would perhaps refrain from that and try to work it out in a separate publication.
  5. There is a clear avenue for connecting the conceptual ideas in the paper with the actual (mathematical) formalism of IIT. Doing this would be beneficial for IIT-affine people.

Minor

1m Abstract: perhaps add “conceivably” here? It really depends on whether you buy into the claim that the phenomenal and the functional are dissociated (e.g. Chalmers’ model)

2m Line 62 “causality and information”. I would delete the information or introduce the term “causal informational structure” instead. It seems that Computational functionalism would agree with the idea that "information is central" to consciousness.

3m Line 213/4 “salient point”, and “a priori perspective”. I did not quite understand whether you mean that there exists a necessary relation between functional and phenomenal accounts  of binding (if P, then F) or whether you just assume this relation as starting point in your theory?

4m In the context of discussing unity/binding it might be good to refer to work that actually tried to give mathematization of these ideas, e.g.

  • Consciousness and Topologically Structured Phenomenal Spaces, Consciousness & Cognition (2019).
  • Kleiner & Ludwig. What is a Mathematical Structure of Conscious Experience?, Synthese (2024).

Round 2

Reviewer 2 Report

Comments and Suggestions for Authors

The authors have responded to all my points and revised the MS where needed. I am happy to endorse the publication of this paper and would look forward to it being published in your journal.